# Changes of mitochondrial ultrastructure and function during ageing in mice and *Drosophila*

**Tobias Brandt[1†], Arnaud Mourier[2,3,4†], Luke S Tain[5], Linda Partridge[5,6], Nils-Göran Larsson[2,7], Werner Kühlbrandt[1\*]**

[1]Department of Structural Biology, Max-Planck-Institute of Biophysics, Frankfurt am Main, Germany; [2]Department of Mitochondrial Biology, Max Planck Institute for Biology of Ageing, Cologne, Germany; [3]Institut de Biochimie et Génétique Cellulaires UMR 5095, Université de Bordeaux, Bordeaux, France; [4]CNRS, Institut de Biochimie et Génétique Cellulaires UMR 5095, Bordeaux, France; [5]Department of Biological Mechanisms of Ageing, Max Planck Institute for Biology of Ageing, Cologne, Germany; [6]Institute of Healthy Ageing, Department of Genetics, Evolution, and Environment, University College London, London, United Kingdom; [7]Department of Medical Biochemistry and Biophysics, Karolinska Institutet, Stockholm, Sweden

**Abstract** Ageing is a progressive decline of intrinsic physiological functions. We examined the impact of ageing on the ultrastructure and function of mitochondria in mouse and fruit flies (*Drosophila melanogaster*) by electron cryo-tomography and respirometry. We discovered distinct age-related changes in both model organisms. Mitochondrial function and ultrastructure are maintained in mouse heart, whereas subpopulations of mitochondria from mouse liver show age-related changes in membrane morphology. Subpopulations of mitochondria from young and old mouse kidney resemble those described for apoptosis. In aged flies, respiratory activity is compromised and the production of peroxide radicals is increased. In about 50% of mitochondria from old flies, the inner membrane organization breaks down. This establishes a clear link between inner membrane architecture and functional decline. Mitochondria were affected by ageing to very different extents, depending on the organism and possibly on the degree to which tissues within the same organism are protected against mitochondrial damage.

**\*For correspondence:** werner. kuehlbrandt@biophys.mpg.de

[†]These authors contributed equally to this work

## Introduction

Mitochondria produce most of the ATP in non-photosynthetic eukaryotes, providing the energy to drive a multitude of cellular processes. Mitochondria have an inner membrane, which surrounds the matrix, and an outer membrane, which surrounds the inner membrane and separates the mitochondrial compartments from the cytoplasm. In addition to a multitude of soluble enzymes and ribosomes, the matrix houses the mitochondrial genome (mtDNA), which in humans and flies encodes 13 hydrophobic subunits of the oxidative phosphorylation system. Thus nearly all mitochondrial proteins are nuclear-encoded and imported into the different mitochondrial compartments by a set of protein translocases (*Pfanner et al., 2014*). The oxidative phosphorylation (OXPHOS) system consists of the respiratory chain complexes I to IV and the mitochondrial $F_1F_o$-ATP synthase, sometimes referred to as complex V. Complexes I to IV transfer electrons from soluble electron donors to molecular oxygen. In this process, complexes I, III and IV generate an electrochemical proton

potential across the mitochondrial inner membrane that is used by the ATP synthase to produce ATP by rotary catalysis (*Leslie et al., 1999*). Oxidative phosphorylation occurs mostly, if not entirely, in the deeply invaginated cristae of the inner mitochondrial membrane (*Gilkerson et al., 2003*; *Vogel et al., 2006*). The ATP synthase forms long rows of dimers at the tightly curved cristae ridges, while the respiratory chain complexes are confined to the remaining membrane regions (*Davies et al., 2012*, *2011*; *Paumard et al., 2002*).

Ageing has long been linked to mitochondrial dysfunction. In mammals, the age-related weakening of physiological functions frequently goes along with a decline in health. While there has been considerable progress in the study of age-associated diseases, the biological mechanisms of ageing remain elusive. Ageing is attributed to either 'programmed' or 'damage-based' processes (*de Magalhaes, 2011*). More than 50 years ago, Harman postulated that ageing results from the accumulation of molecular damage caused by oxygen radicals, often called reactive oxygen species (ROS) (*Harman, 1956*). Oxygen radicals are side products of electron transfer reactions in respiratory chain complexes I and III (*Dröse and Brandt, 2012*). Later refinements of Harman's free radical theory of ageing postulated that an impairment of respiratory chain function increases ROS production, resulting in greater mtDNA damage, which in turn would compromise the turnover of damaged respiratory chain complexes, resulting in a vicious circle of damage and decline (*Harman, 1972*). The mitochondrial DNA mutator mouse model has provided experimental evidence of a causative link between mtDNA mutations and an ageing phenotype in mammals (*Trifunovic et al., 2004*), although mitochondrial defects in this model were found not to be associated with an increase in ROS production (*Trifunovic et al., 2005*). Several other key predictions of the free-radical theory (*Stuart et al., 2014*) have also not been substantiated. For example, in some model organisms, such as the nematode *Caenorhabditis elegans*, moderately elevated levels of ROS, induced either chemically or through genetically engineered defects in the respiratory chain, actually increase lifespan (*Heidler et al., 2010*; *Lee et al., 2010*; *Yang and Hekimi, 2010*). In the same organism, removal of the ROS-scavenging superoxide dismutase (SOD) does not increase oxidative damage to mtDNA and has no apparent effect on lifespan (*Doonan et al., 2008*; *Gruber et al., 2011*; *Van Raamsdonk and Hekimi, 2009*). In the fly, SOD knockouts do have a decreased lifespan, but they do not accumulate mtDNA mutations more quickly than wildtype (*Itsara et al., 2014*).

In addition to energy conversion and ATP production, the oxidative phosphorylation system also has a key role in shaping the inner membrane cristae (*Davies et al., 2012*, *2011*; *Paumard et al., 2002*). Accumulating oxidative damage and OXPHOS dysfunction might, therefore, be expected to affect mitochondrial ultrastructure, which should be visible by electron microscopy. Recent analyses of yeast and mouse mitochondria show that a defect in the supra-molecular organisation of the ATP synthase results in aberrant cristae morphology (*Davies et al., 2012*; *Mourier et al., 2014*). In aged cultures of the short-lived filamentous fungus *Podospora anserina*, the inner membrane strikingly vesiculates and ATP synthase dimers break down (*Brust et al., 2010*; *Daum et al., 2013*). However, little is known about the impact of ageing on mitochondrial morphology and membrane structure in metazoans. Previous studies of the ultrastructure of mitochondria in mouse liver by electron microscopy of thin plastic sections revealed anomalous cristae in a subpopulation of the organelles (*Wilson and Franks, 1975*). A study of mouse skeletal muscle mitochondria found that mitochondria from animals that had exercised in an electrically-driven treadmill occasionally lost their cristae, whereas no differences were observed in non-trained animals (*Ludatscher et al., 1983*). Mitochondria from aged *D. melanogaster* flight muscle were found to have cristae 'swirls' that were attributed to oxidative damage (*Walker and Benzer, 2004*), and mitochondria in aged *Drosophila repleta* heart muscle were enlarged (*Sohal, 1970*).

We chose *D. melanogaster* (average lifespan >50 days depending on environmental conditions [*Linford et al., 2013*]) and mouse (average life span 107 weeks for females and 114 weeks for males [*Turturro et al., 2002*]) as two well-established metazoan ageing models (*Cho et al., 2011*) for a systematic study of mitochondrial ultrastructure, respiration and ROS production in young and old animals. Our results reveal clear tissue- and organism-related age-specific differences, establishing an apparent link between mitochondrial ultrastructure and function.

## Results

### Effects of ageing on the function and ultrastructure of mouse mitochondria

#### Mitochondrial OXPHOS activity and ROS homeostasis

To find out how ageing impacts mitochondrial function and ultrastructure, mitochondria from heart, liver and kidney of young (20 weeks) and old (80–96 weeks) mice were isolated for electron cryo-tomography (cryoET) and high-resolution respirometry. Isolated mitochondria were incubated with respiratory substrates that deliver electrons at the level of complex I (pyruvate, glutamate and malate) or complex II (succinate and rotenone). The mitochondrial oxygen consumption was recorded under phosphorylating (addition of ADP and P$_i$; state 3), non-phosphorylating (addition of oligomycin to inhibit ATP synthase; state 4) and uncoupled conditions (addition of CCCP). Mitochondrial respiratory rates normalised to protein content were found to be highest in heart mitochondria, ~2–3 fold lower in kidney and ~5 fold lower in liver (*Figure 1A*). In line with previous reports (*Mulligan et al., 2014; Weindruch et al., 1980*; *Wilson et al., 1975*), our investigation showed no significant differences in respiratory rates between mitochondria isolated from 20-week-old and 80-week-old heart tissue (*Figure 1A*), and no age-related change in the rate of ATP production was observed (*Figure 1C*). Moreover, maximal activities of key mitochondrial enzymes remained unchanged (*Figure 1E*). These results prompted us to investigate how ageing impacts mitochondrial ROS homeostasis. We first assessed the peroxide yield, defined as the hydrogen production rate normalized to the mitochondrial respiration assessed under the same conditions (*Votyakova and Reynolds, 2001*). Surprisingly, no age-dependent increase in hydrogen peroxide release relative to the amount of oxygen consumed was observed in mitoch ondria isolated from heart, liver or kidney (*Figure 1B*). To investigate ROS homeostasis further, we analysed the steady-state levels and activity of the antioxidant enzymes superoxide dismutase (SOD1, SOD2) and catalase in mitochondria from young and old heart, liver or kidney by quantitative western blot electrophoresis and gel densitometry (*Figure 1D*). This revealed a tendency towards lower levels of antioxidant enzymes in aged mice. Catalase activity was reduced by 30% in old mouse liver (*Figure 1F*) and SOD activity was almost halved in old mouse kidney (*Figure 1—figure supplement 1*).

#### Tomography of mouse heart mitochondria

Vitrified samples of the mitochondrial preparations used for high-resolution respirometry were analysed by electron cryo-tomography (cryo-ET). Mitochondria isolated from young mouse heart showed morphologies typical of tissues with a high energy demand. Stacks of parallel, thin and flat lamellar cristae were embedded in a dense matrix (*Figure 2*, left, *Video 1*). Cristae frequently appeared discontinuous in 2D slices, but 3D volumes indicated that this was due to fenestration of the lamellar disks rather than to disconnected cristae vesicles (*Figure 2—figure supplement 1A*). Crista junctions were circular (70%, average diameter 15 ± 2 nm) or slightly elongated (30%, 16 ± 1 × 29 ± 2 nm, dimensions ± standard deviations measured in the typical tomogram of *Figure 2*, top row, second from left). Most cristae were branched and connected to the inter-membrane space by more than one junction. They were also highly interconnected through narrow apertures (*Figure 2—figure supplement 1B*). In 20% of the mitochondria examined, membranes of two or more neighbouring cristae were so closely appressed that there was almost no matrix between them. These regions often showed membrane 'swirls' of high and variable membrane curvature, involving several neighbouring crista lamellae (*Figure 2—figure supplement 1C*). There was no apparent difference between the structures of isolated sub-sarcolemmal and interfibrillar mitochondria. Comparing mitochondria from young and old mouse hearts indicated similar morphologies (*Figure 2*, right), except that about 24% of mitochondria from aged heart tissue had some exceptionally wide cristae (*Table 1*; *Figure 2—figure supplement 1D*).

#### Tomography of mouse liver mitochondria

The morphology of mitochondria isolated from mouse liver was conspicuously different from that of mouse heart (*Figure 3*, left; *Video 2*). The cristae were more heterogeneous, less regular and not arranged in parallel stacks. They were generally wider and did not span the entire mitochondrion. Also, the cristae were less interconnected to one another and to the inner boundary membrane. As

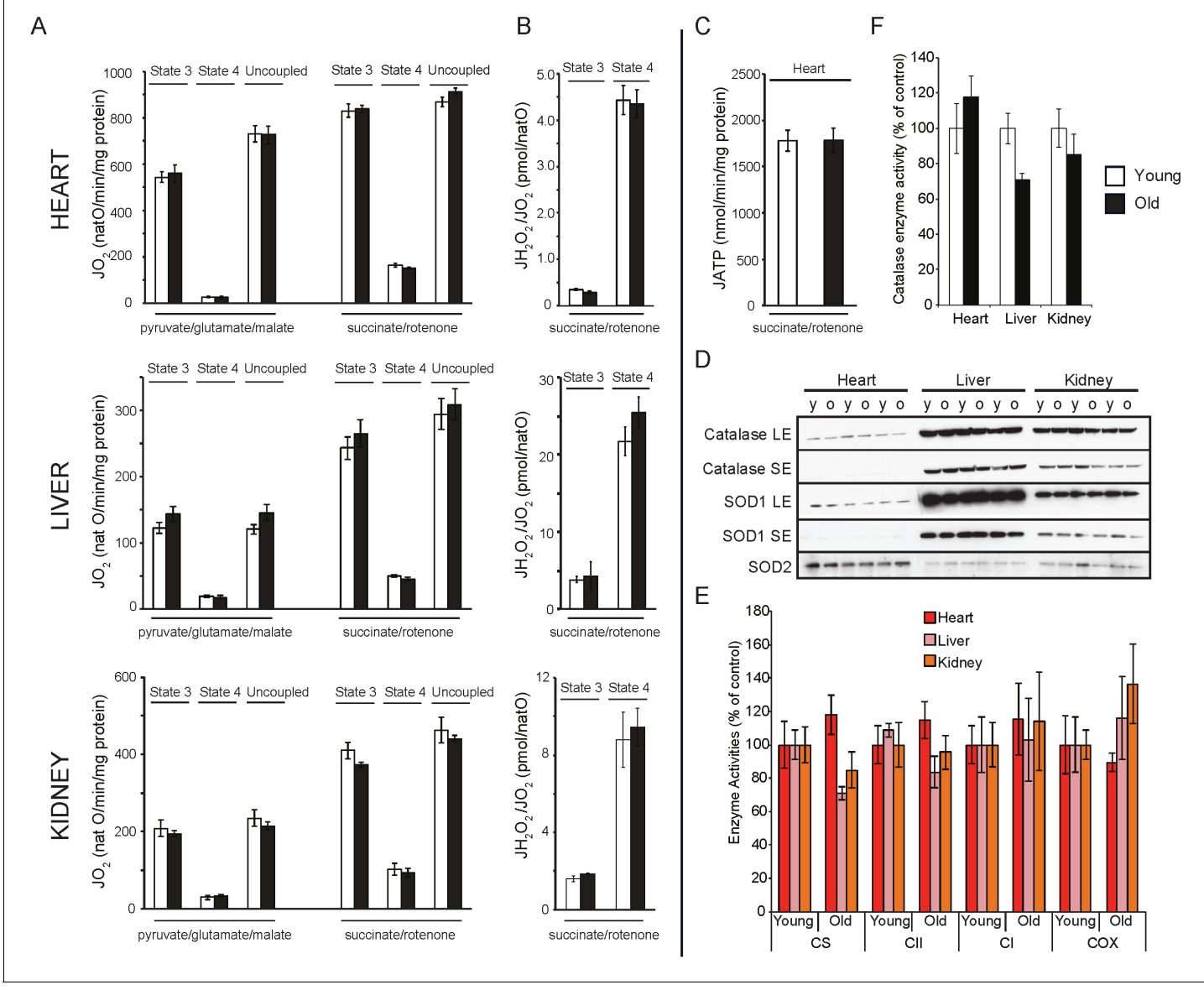

**Figure 1.** Bioenergetic and functional analysis of mitochondria from heart, liver and kidney from young (20 weeks old; white bars) and old (80–96 weeks old; black bars) mice. (**A**) Oxygen consumption rate of mitochondria isolated from young (n = 7–8) or old (n = 8) animals. Isolated mitochondria were incubated with electron donors to complex I (pyruvate, glutamate, malate) or complex II (succinate, complex I inhibited with rotenone). Each set of substrates was successively combined with ADP (to assess the phosphorylating respiration, state 3), oligomycin (to measure non-phosphorylating respiration, state 4) and finally uncoupled by adding increasing concentrations of CCCP. (**B**) Mitochondrial peroxide yield assessed in mitochondria from young (n = 3–4) and old (n = 3–4) animals. (**C**) Mitochondrial ATP synthesis rate in heart mitochondria from young (n = 4) and old (n = 3) animals. Error bars indicate mean ± standard error of the mean (SEM). (**D**) Steady-state levels of different antioxidant enzymes in heart, liver and kidney mitochondria isolated from young (y) and old (o) mice were quantified by western blot analyses. Long (LE) and short exposure (SE) times are presented for catalase and SOD1 detection. (**E**) Citrate synthase (CS) and respiratory chain enzyme activity (complex I, II and IV) measurements in heart, liver and kidney tissue extracts from young (n = 4) and old (n = 4) animals. Error bars indicate mean ± SEM. (**F**) Catalase enzyme activity measured in heart, liver and kidney tissue extracts from young (n = 4) and old (n = 4) animals.

The following figure supplement is available for figure 1:

**Figure supplement 1.** Steady-state levels of different antioxidant enzymes in heart, liver and kidney extracts, as assessed by quantitative densitometry of western blots in *Figure 1D*.

## Young

## Old

**Figure 2.** Cryo-ET of heart mitochondria from young (left, 20 weeks old) and old (right, 80–96 weeks old) mice. Upper panel: slices through tomographic volumes (scale bars, 250 nm). Lower panels: segmented 3D volumes of two typical mitochondria with closely stacked, roughly parallel cristae (blue). The outer membrane (omitted for clarity in the right panel) is yellowish grey. Cristae are connected to the intermembrane space by well-defined, multiple crista junctions.

The following figure supplement is available for figure 2:

**Figure supplement 1.** Detailed views of cristae in young and old mouse heart mitochondria.

in cardiac mitochondria, fenestration was observed, although it was less frequent. The matrix was very dense, making it difficult to segment larger mitochondria. In one sample, the matrix contained dense granules up to 50 nm in diameter (*Video 2*). No such granules were found in heart mitochondria. In two instances, mitochondria had central low-density compartments that we refer to as voids (see below).

Tomograms of mouse liver mitochondria from aged animals revealed two different phenotypes. While the majority of mitochondria looked similar to those isolated from young mouse liver, 32% (n = 31) had conspicuous low-density compartments, or voids, in the centre of the organelle (*Figure 3*, right, *Video 3*; *Table 1*). Segmentation of 3D volumes revealed that the membrane

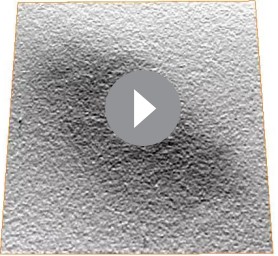

**Video 1.** Tomographic volume and 3D segmentation of the mitochondrion from young mouse heart shown in *Figure 2*, upper left, right hand panel.

**Table 1.** Overview of organisms and tissues analyzed. #number of animals dissected (mouse) or number of mitochondrial preparations (fly) vs number of individual mitochondria examined tomographically.

| Organism | Age | Tissue | Samples / mitochondria[#] | Mitochondria with abnormal morphology (%) |
|---|---|---|---|---|
| mouse | young | heart | 4/27 | wide cristae 4%; cristae membrane swirls 19% |
| | old | heart | 3/17 | wide cristae 24%; cristae membrane swirls 6% |
| | young | liver | 5/18 | voids 11%; granules 6% |
| | old | liver | 4/31 | voids 32%; granules 6%; apoptotic 3% |
| | young | kidney | 3/33 | apoptotic 12%; granules 3% |
| | old | kidney | 2/22 | apoptotic 18%; granules 18% |
| | mutator | heart | 2/10 | membrane swirls 30%; membrane enclosures 40%; granules 20% |
| | mutator | liver | 2/10 | voids 40%; granules 20% |
| fly | young | whole organism | 3/29 | elongated (axial ratio > 3) 10%; wide cristae 10% |
| | old | whole organism | 3/39 | elongated (axial ratio > 3) 18%; wide cristae 15%; branched 5%; various other 23% |

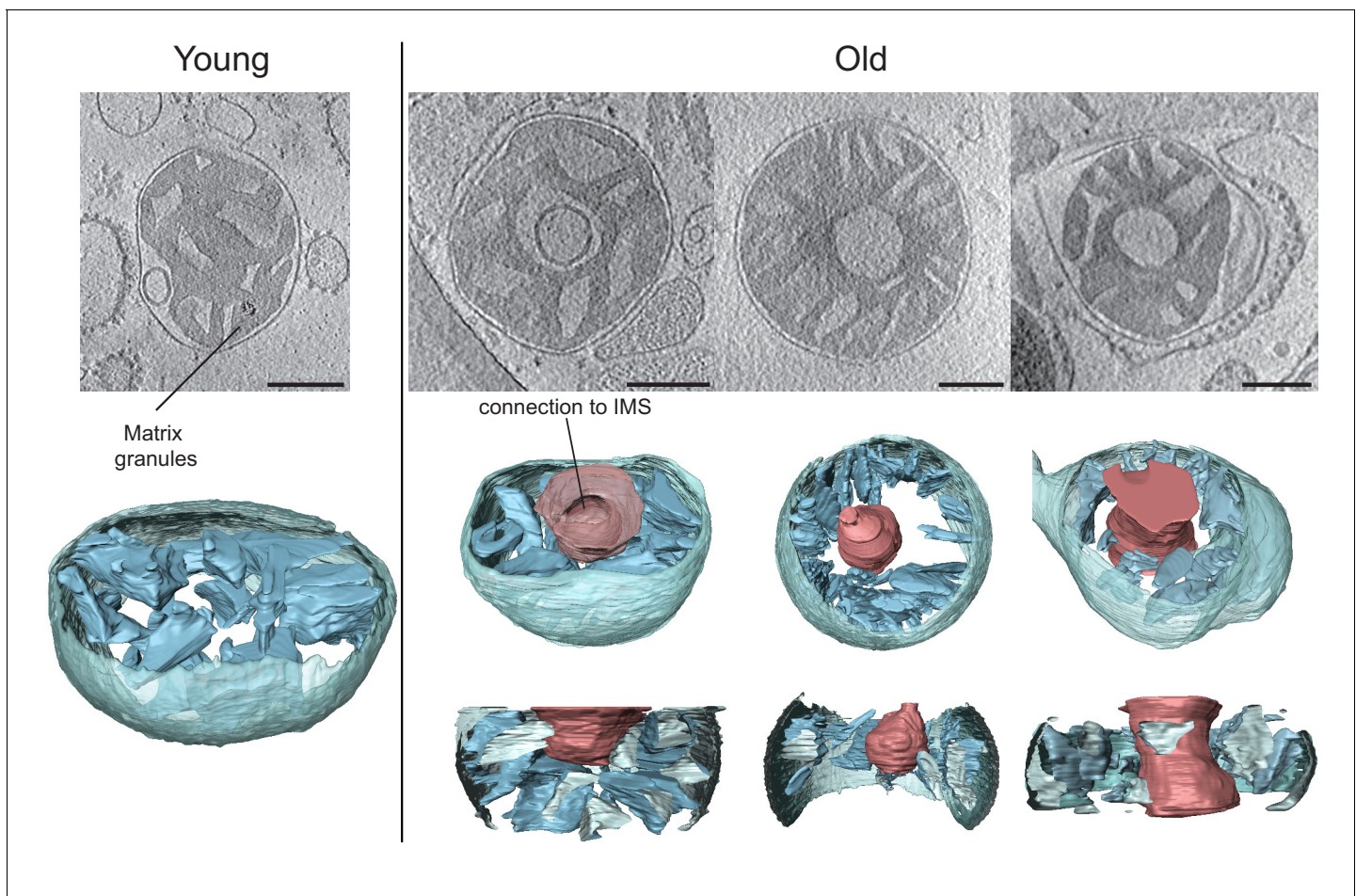

**Figure 3.** Cryo-ET of liver mitochondria from young (left, 20 weeks old) and old (right, 80 weeks old) mice. Upper panel: slices through tomographic volumes (scale bars, 250 nm). Lower panels: segmented 3D volumes. About 25% of the mitochondria from old animals have large central voids (red). The voids were connected to the inter-membrane space (IMS) by openings of variable size.

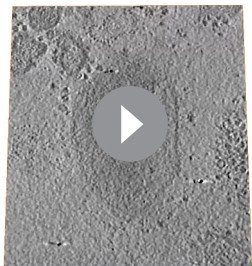

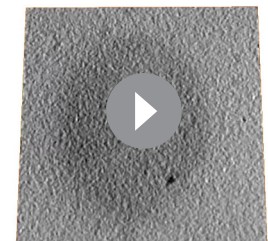

**Video 2.** Tomographic volume and 3D segmentation of the mitochondrion from young mouse liver shown in *Figure 3*, upper left.

**Video 3.** Tomographic volume and 3D segmentation of the mitochondrion from old mouse liver shown in *Figure 3*, upper right, central panel.

delineating these central voids was continuous with the inner membrane. In three segmented mitochondria, the voids accounted on average for 22% of the total cristae surface. To estimate their volumes, voids and mitochondria were approximated as simple geometrical shapes (spheres or cylinders for voids, cylinders for mitochondria). In the three segmented organelles, the voids occupied on average 6% of the total mitochondrial volume. The voids were not internal vesicles, but were connected to the inter-membrane space via apertures of varying size. In some cases two such connections were observed on opposite ends of the voids, resulting in a toroid or doughnut-shaped matrix. In other cases, the outer membrane appeared to protrude into the inner membrane voids. The cristae extending from the boundary membrane into the matrix looked normal, but none protruded into the matrix from the membrane defining the voids. In two tomograms, dense granules measuring up to 100 nm in diameter were observed in the matrix.

## Tomography of mouse kidney mitochondria

Kidney mitochondria resembled heart mitochondria more closely than those from liver (*Figure 4A*). Mitochondria from young and old mouse kidney were similar, except that dense matrix granules were found in 18% of samples from old animals compared to only one (3%) found in the samples from young animals. Cristae were lamellar but generally wider than those in cardiac tissue and less stringently arranged in parallel stacks. Occasionally cristae formed membrane swirls. 18% of kidney mitochondria from old mice and 12% from young mice had the characteristic morphology described for apoptotic cells (*Scorrano et al., 2002*; *Sun et al., 2007*). In these mitochondria, the cristae were irregular, not arranged in any discernible pattern, and the membrane curvature was locally inverted. Crista junctions were very wide or not discernible, resembling those of prohibitin-deficient mouse mitochondria, in which OPA1 (which is essential for cristae junction formation) is incorrectly processed (*Merkwirth et al., 2008*). The inner membrane enclosed a convoluted but apparently continuous volume. The width of the inter-membrane space between the inner boundary and outer membrane was largely the same as in normal heart, liver or kidney (*Figure 4B*), indicating that the mitochondria were intact and had not suffered from osmotic shock during isolation (*Wrogemann et al., 1985*). Results are summarized in *Table 1*.

## Mutator mouse mitochondria

Next, we compared the function and ultrastructure of wild-type mouse mitochondria to those from the heart and liver of mtDNA mutator mice, a strain with a premature ageing phenotype (*Trifunovic et al., 2004*). A minor respiratory defect was found under phosphorylating and uncoupling conditions in heart from mtDNA mutator mice (*Figure 5A*), consistent with previous observations (*Trifunovic et al., 2004*, *2005*). The morphology of heart mitochondria from the mtDNA mutator mice resembled wildtype heart mitochondria, except for a high incidence (40%) of

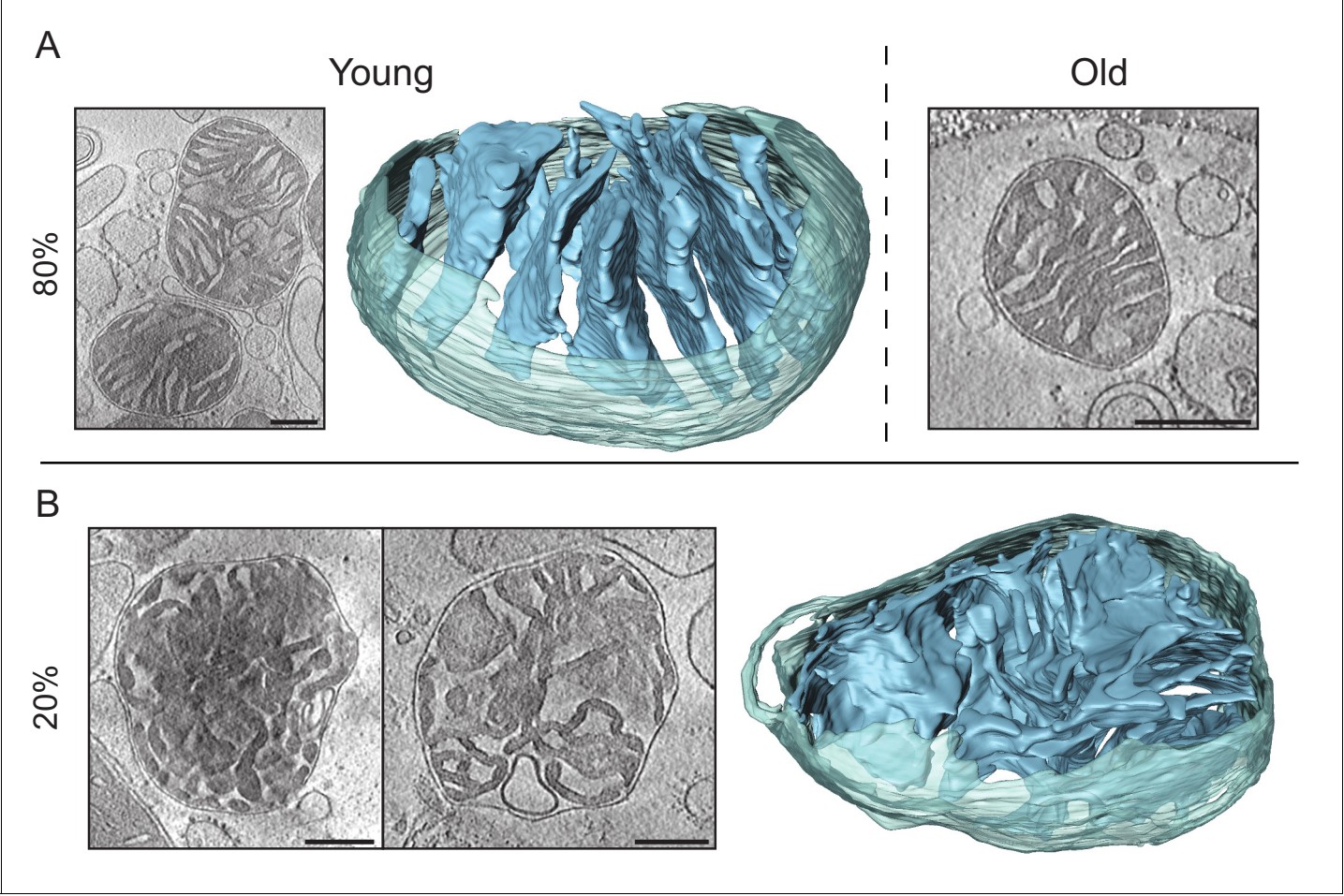

**Figure 4.** Cryo-ET of kidney mitochondria from young (20 weeks old) and old (80 weeks old) mice. (**A**) About 80% of the kidney mitochondria from young (left) or old (right) animals had lamellar and locally parallel cristae, not unlike heart mitochondria, except that the cristae were less tightly packed. (**B**) About 20% of the mitochondria from young and old kidney showed an inner membrane morphology resembling that in apoptotic cells (*Scorrano et al., 2002*), without any discernible pattern and wide or irregular junctions. Scale bars, 250 nm.

ellipsoidal peripheral voids connected to the cristae or the inter-membrane space that were delineated by a double membrane (*Figure 5B*). The matrix space between the two membranes was minimal. In one mtDNA mutator mouse heart mitochondrion, the outer membrane was punctured by small openings (*Figure 5B*, left).

Mitochondria isolated from mtDNA mutator mouse liver showed the same central low-density voids (*Figure 5C*) as liver mitochondria from old wild-type mice (*Figure 3*), but at greater prevalence (40%; *Table 1*). In one case, cristae with apparently normal junctions extended from the central void into the matrix, which was not observed in wild-type mice (*Figure 5C*). About 20% of mtDNA mutator mouse liver mitochondria contained dense matrix granules (*Table 1*).

### *D. melanogaster* mitochondria show profound age-associated changes

We also examined the ultrastructure and function of mitochondria from young and old *D. melanogaster*, a non-mammalian metazoan. The flies showed standard sigmoidal survival curves, with a mean lifespan of 69.5 days and a maximum lifespan of 78.5 days (*Figure 6A*). Life span correlated with climbing ability, as reported previously (*Rhodenizer et al., 2008*). Climbing ability remained constant for 25 days and then dropped rapidly (*Figure 6A*). We investigated the respiratory activity of mitochondria from young and old flies (15 and 70 days, respectively; *Figure 6A*). The respiration rate in old flies decreased by up to 60% (*Figure 6B*), and the peroxide yield increased by

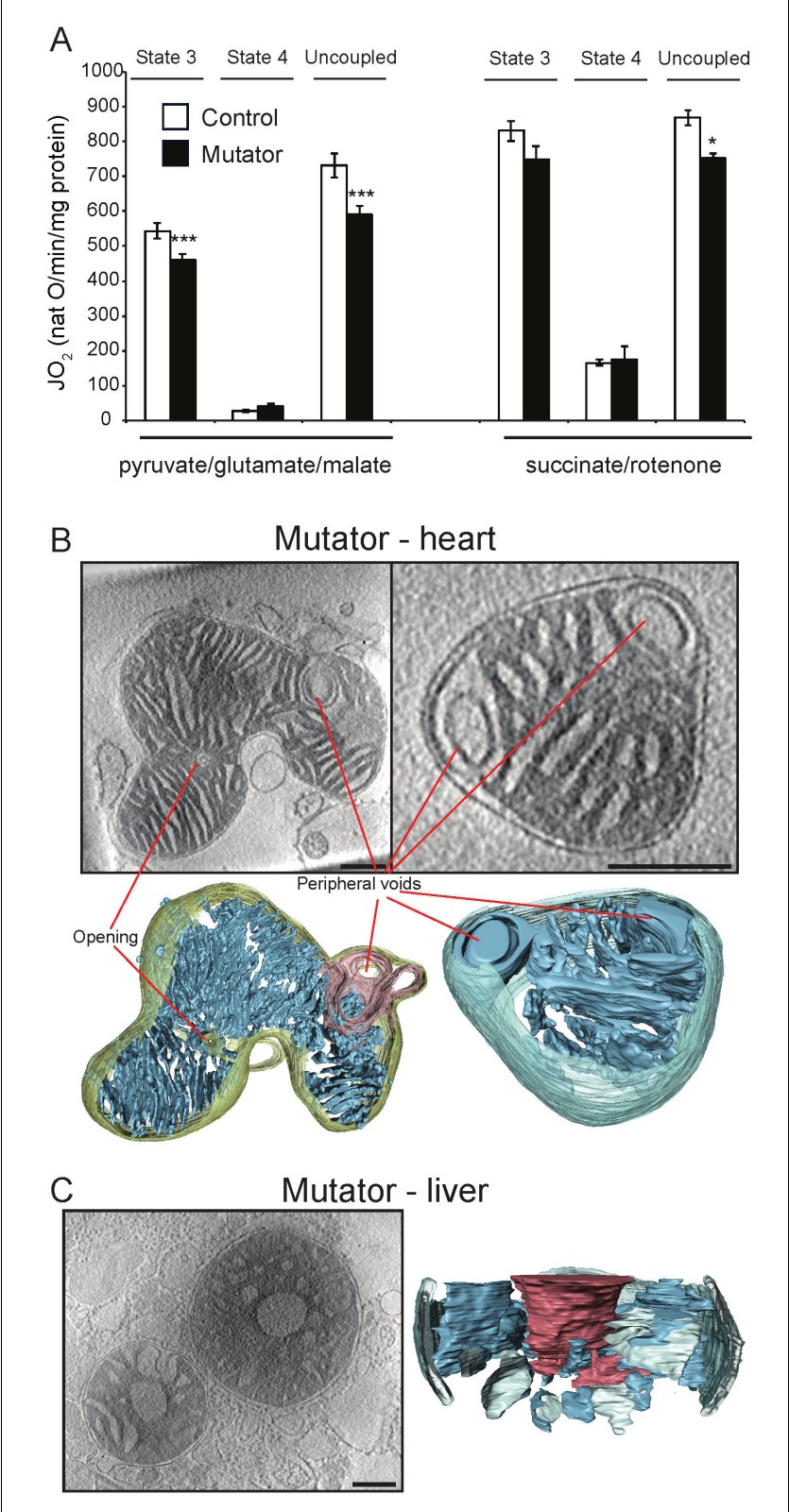

**Figure 5.** Activity and ultrastructure of mtDNA mutator mouse mitochondria. (**A**) Oxygen consumption rate assessed in heart mitochondria from control (white bars, n = 8, 30 weeks old) and mutator (black bars, n = 8, 30 weeks old) mice. Mitochondria were isolated and analysed as in *Figure 1*. (**B–C**) Cryo-ET. (**B**) Mutator mouse heart mitochondria had lamellar, parallel cristae that were similar to those of wild-type heart mitochondria (see *Figure 2*), *Figure 5 continued on next page*

*Figure 5 continued*

but with occasional peripheral voids at the inner boundary membrane. (**C**) About 40% of liver mitochondria from mutator mice had low-density central voids, as in old liver (see *Figure 3*).

40–100% (*Figure 6C*), indicating that mitochondrial function is severely compromised, confirming an earlier report (*Ferguson et al., 2005*).

We performed cryo-ET on three sample preparations per age group to find out whether and how this functional decline is reflected in mitochondrial morphology. Results are summarized in *Table 1*. Mitochondria from young flies (n = 29; *Figure 6D*, *Video 4*) looked similar to those from mouse heart, with a dense matrix and thin, lamellar, mostly parallel cristae. All cristae had ridges indicative of ATP synthase dimer rows (*Davies et al., 2011*; *Strauss et al., 2008*) and several junctions to the intermembrane space. Wide cristae were observed in 10% of the mitochondria, similar to mouse heart. A small subpopulation of mitochondria (10%) was unusually elongated (axial ratio above 3:1) but looked otherwise normal. By contrast, the structure of mitochondria from old *D. melanogaster* (n = 39) was heterogeneous (*Figure 6E*). Approximately 75% of mitochondria had normal, well-developed cristae, although in 15% the cristae were wider (not shown). Some of these mitochondria were unusually long and thin (18%), or branched (5%) (*Figure 6E*, top left). In the remaining 25% of old fly mitochondria, the inner membrane assumed a variety of non-standard shapes. Several organelles that were identified as mitochondria by their characteristic double membrane and dense matrix appeared to lack cristae entirely (*Figure 6E*, top right) or the cristae were minimally developed (10%). Many cristae were not connected to the intermembrane space and were, therefore, small vesicles completely surrounded by matrix (*Figure 6E*, lower right, *Video 5*). In one mitochondrion, the cristae were spherical (*Figure 6E*, centre left). In two others, they were concentric (*Figure 6E*, lower left, *Video 6*), lacking the membrane ridges associated with ATP synthase dimer rows (*Davies et al., 2011*; *Strauss et al., 2008*). These observations establish a strong link between mitochondrial inner membrane organisation and respiratory function in *D. melanogaster*.

## Discussion

Our study provides a systematic assessment of changes in mitochondrial function and inner membrane structure upon ageing in two common metazoan ageing model organisms, mouse and *D. melanogaster*. 3D volumes of entire mitochondria at an estimated resolution of 3 to 5 nm were generated by cryo-ET. As far as possible, mitochondria from different organisms and tissues were treated in the same way. The fact that there are clear differences between mitochondria from young and old mouse liver, for example, and that these mitochondria look different from heart and kidney mitochondria allows us to conclude that the isolation process itself does not affect mitochondrial membrane structure significantly. We conclude further that the different appearance of the young mouse liver mitochondria is not an artefact of isolation, but a genuine feature.

We found a clear correlation between OXPHOS capacity and the number of inner membrane cristae per unit volume. Mitochondrial respiration was highest in mouse heart and young fly mitochondria, which were entirely filled with closely stacked, parallel cristae. This arrangement, which is characteristic for tissues with high respiratory activity, maximises the membrane area available for oxidative phosphorylation (*Davies et al., 2011*). In mouse heart and young fly mitochondria, we frequently observed cristae fenestration, which is a morphological characteristic of tissues with high energy demand (*Slautterback, 1965*; *Smith, 1963*). Fenestration increases the total length of cristae ridges that harbour the ATP synthase, and hence the potential for ATP production. By comparison, mouse liver mitochondria had fewer cristae and the matrix was both denser and more voluminous, in line with the lower respiratory rate and higher metabolic activity of liver cells. Mouse kidney mitochondria were structurally more diverse. Remarkably, a significant proportion of kidney mitochondria from both young and old mice had a morphology typical of apoptotic cells (*Scorrano et al., 2002*; *Sun et al., 2007*). This correlates well with the recently reported continuous turnover and short lifespan of kidney cells of about 30–60 days (*Rinkevich et al., 2014*), compared to the 200–400 day lifespan of rat hepatic cells (*Macdonald, 1961*) and the very low turnover rates for cardiomyocytes of 1.3–4% per year (*Malliaras et al., 2013*). The peroxide yield correlated inversely with respiratory

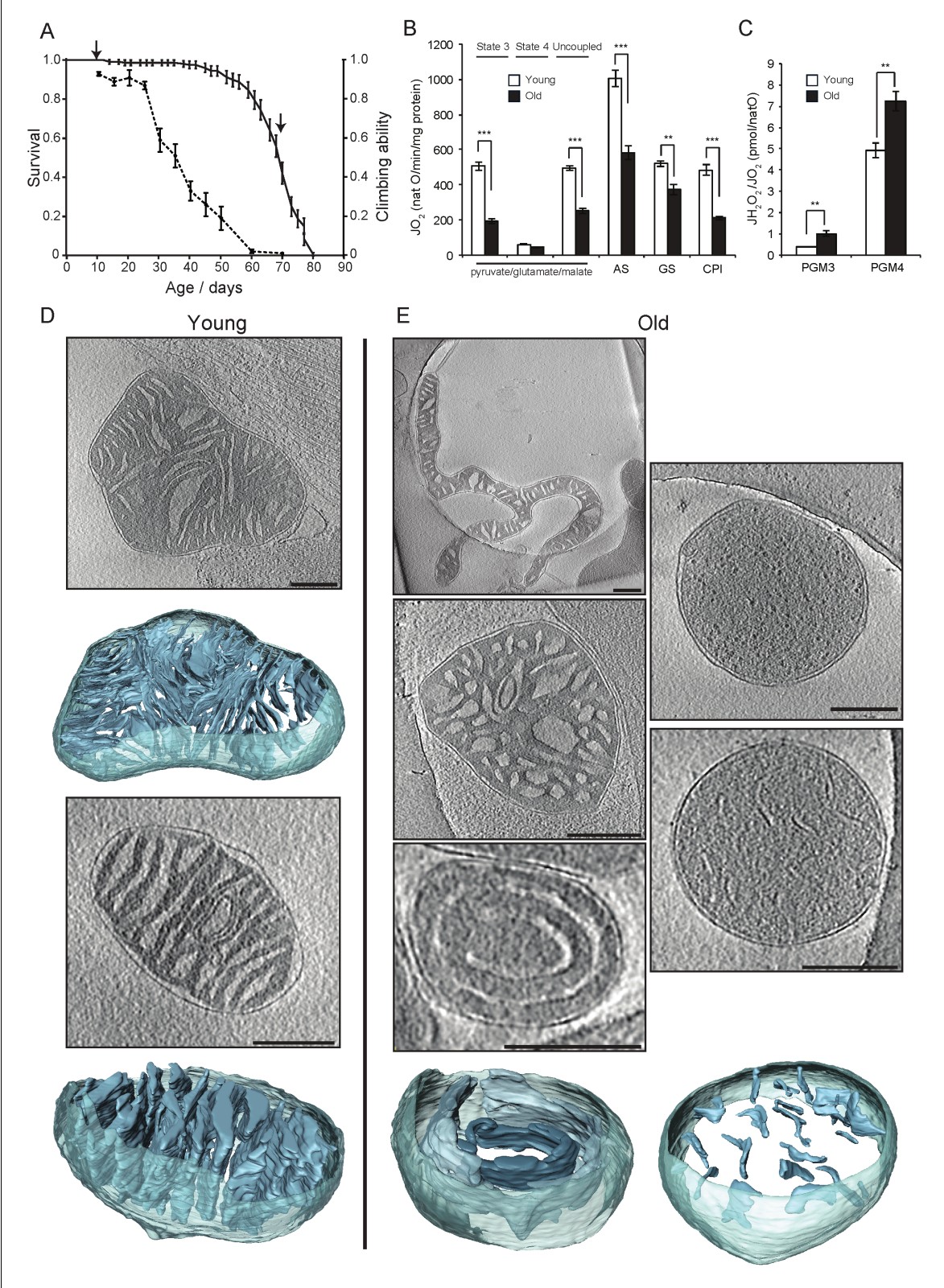

**Figure 6.** Activity and ultrastructure of *D. melanogaster* mitochondria. (**A**) Survival rates (n = 150; solid line) and climbing ability (n ≥ 25; dashed line) of $w^{Dah}$ wild-type flies. Error bars represent SEM and arrows indicate sampling points for young (15 days old) and old (70 days old) flies. (**B**) Oxygen consumption rate assessed in mitochondria from young (white bars, n = 3) and old (black bars, n = 3) flies. Mitochondria were isolated and analysed as in *Figure 1*. Succinate and glycerol-3-phosphate (AS), and finally rotenone were added (GS) for comparison to complex I-driven respiration (CPI). (**C**)
*Figure 6 continued on next page*

*Figure 6 continued*

Mitochondrial peroxide yield in mitochondria from young (white bars, n = 3) and old (black bars, n = 3) flies. (D–E) Cryo-ET of typical mitochondria from young (D) or old (E) flies. Mitochondria from young flies had lamellar, mostly parallel cristae, similar to those from mouse heart. Mitochondria from old flies had highly variable shapes and cristae organisation, with the following main types (clockwise from top left): elongated and branched morphology; round, lacking cristae; small disconnected cristae; concentric narrow cristae (note that the innermost vesicle, shown in a darker shade of blue, appears to be unconnected to other membranes); irregular, wide cristae.

activity and was highest in mouse liver and lowest in mouse heart, suggesting that liver mitochondria may be affected by oxidative damage more severely than mitochondria from cardiac tissue.

Unexpectedly, levels of mitochondrial respiration, respiratory chain activity, hydrogen peroxide production or steady-state levels of antioxidant enzymes did not vary greatly with age, indicating moderate changes at most in respiratory rates and hydrogen peroxide yield in ageing mice. This contravenes the free-radical theory of ageing, which postulates a major age-related increase in ROS production and respiratory defects. In line with our results on mitochondrial function, the morphology of mouse heart mitochondria did not change much with age. By contrast, one out of four mitochondria from old mouse livers had a striking, age-specific phenotype characterized by a central matrix void. We found the same feature in mitochondria of mtDNA mutator mouse livers, where it was considerably more prevalent. Although we do not yet know what causes the voids and how they affect mitochondrial function, they evidently reflect a specific reorganisation rather than a random breakdown of the inner membrane. The lack of sharp cristae ridges and crista junctions in the void membranes suggests that these mebranes do not contain ATP synthase dimers and hence do not contribute to oxidative phosphorylation. The fact that mitochondria with these central voids have otherwise normal cristae may explain why no age-related reduction in overall respiratory activity was evident. If the majority of mitochondria are normal, they can apparently compensate for a loss of inner membrane area available for oxidative phosphorylation in 25% of the total population. Similar ring- or cup-shaped mitochondria with central cavities have been reported (*Ghadially, 1988*) and ascribed to the effects of drugs, toxins (*David and Kettler, 1961*; *Stephens and Bils, 1965*) or oxidative damage (*Ding et al., 2012*). Although the cavities found in these earlier studies were lined by both the outer and inner mitochondrial membranes, the central voids described here may likewise result from such damage, as damage to mtDNA is known to accumulate in liver tissues (*Barazzoni et al., 2000*; *Yen et al., 1991*). If denatured respiratory chain complexes are cleared from the membrane, the lipids left behind would be expected to form such featureless membrane regions expanding into the matrix interior. The matrix granules that we found in old liver and kidney mitochondria may consist at least in part of denatured respiratory chain complexes, consistent with their absence from old heart mitochondria, in which we did not observe matrix granules. Indeed, it has been shown that these granules contain complex IV subunits (*Hertsens et al., 1986*) in addition to lipids, glycoproteins and calcium (*Ghadially, 2001*; *Jacob et al., 1994*).

Interestingly, changes in the membrane structure of mouse mitochondria were organ-dependent, in a way that suggests different degrees of resilience against ageing. Mouse cardiac mitochondria showed the highest respiratory rates but appeared to be protected most effectively from oxidative damage, as indicated by their low and unchanged peroxide yield. Again, these findings contradict the classical free-radical theory, which would predict that cardiomyocytes with their high density of highly active mitochondria produce more ROS and thus age faster (*Stuart et al., 2014*). An analysis of DNA methylation has shown that heart tissue ages more slowly than would be predicted chronologically

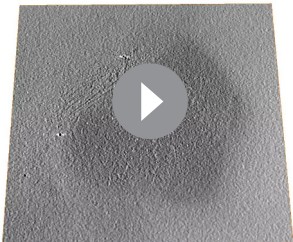

**Video 4.** Tomographic volume and 3D segmentation of the mitochondrion from young fly shown in *Figure 6D*, upper panel.

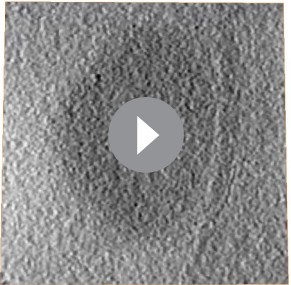

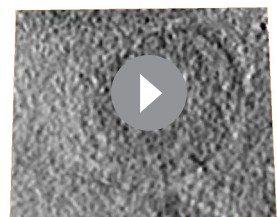

**Video 5.** Tomographic volume and 3D segmentation of the mitochondrion from old fly shown in *Figure 6E*, lower right.

**Video 6.** Tomographic volume and 3D segmentation of the mitochondrion from old fly shown in *Figure 6E*, lower left.

(*Horvath, 2013*). This observation is in line with our findings, which thus imply that mitochondria in vital organs with slow regeneration, such as the heart, are protected from damage more effectively than mitochondria in other organs. The effect may be genetically controlled through the expression levels of superoxide dismutase or catalase. Overexpression of catalase targeted to mitochondria has been shown to attenuate ageing effects in the murine heart (*Dai et al., 2009*). However, ROS signalling (*Lenaz, 2012*) requires a fine equilibrium of ROS production and sequestration that is probably tissue-dependent.

The most striking differences in our study, both with respect to morphology and activity, were found between mitochondria from young and old *D. melanogaster*. About half of the mitochondria from old flies had lost the standard organization of the inner membrane into boundary membranes with well-developed cristae and crista junctions. Many had round or concentric cristae that lacked sharp ridges. In extreme cases, cristae or crista junctions were completely absent. Measurements of respiratory activity in old fly mitochondria indicated that oxygen uptake was decreased by a factor of almost two and peroxide yield was increased by more than 50%, in line with increased mitochondrial $H_2O_2$ production in live ageing *Drosophila* (*Cochemé et al., 2011*). These results suggest strongly that about 50% of the old fly mitochondria are inactive, consistent with the observation that about 25% of the mitochondria in old flies lack normal cristae or crista junctions and that an additional 18% deviate from the standard morphology. Such a drastic breakdown of mitochondrial structure and function would result in the death of the organism within a short period.

It is interesting to compare our results on mouse and fruit fly mitochondria to similar studies on the filamentous fungus *P. anserina* (*Brust et al., 2010*; *Daum et al., 2013*), another well-characterized ageing model (*Scheckhuber and Osiewacz, 2008*). *P. anserina* has an average lifespan of only 18 days, three times shorter than *D. melanogaster,* and almost 50 times shorter than mouse. Morphological changes in *P. anserina* mitochondria were both more homogenous and more extreme than those in fruit flies, affecting about 80% of organelles from senescent populations (*Brust et al., 2010*). Cryo-ET of aged *P. anserina* mitochondria or inner membrane vesicles indicated that the cristae had receded into the inner boundary membrane and that ATP synthase dimers dissociated into monomers (*Daum et al., 2013*). In terms of inner membrane morphology, old *P. anserina* mitochondria resembled the quasi-apoptotic subpopulation in mouse kidney, the tissue with the highest turnover rate in our study, suggesting similar mechanisms of programmed ageing. On the one hand, the higher proportion of functional mitochondria in old flies indicates that the decline is less complete and slower than that in *P. anserina*. On the other hand, it is much more rapid in *D. melanogaster* than in mouse, suggesting a link between the complexity of an organism and the rate of ageing.

## Conclusions

The increasing complexity of organisms goes along with an increasingly complex ageing process. For the primitive multicellular eukaryote *P. anserina*, a straightforward correspondence between

age, mitochondrial ultrastructure and organization of the mitochondrial ATP synthase has been shown (*Brust et al., 2010*; *Daum et al., 2013*). In *Drosophila*, we now establish a link between inner membrane morphology and functionality, which correlates closely with age and agility. In mouse, the relationship between inner membrane ultrastructure, function and age is less clear-cut and evidently tissue-dependent. While mouse heart mitochondria show little if any change with age, a quarter of liver mitochondria display a severe, age-related phenotype that does not seem to result in an overall reduction of oxidative phosphorylation. A relatively high proportion of kidney mitochondria in young and old mice resemble those observed during apoptosis, consistent with the high turnover of kidney cells. Our data thus indicate major differences in how ageing relates to mitochondrial morphology and function in metazoans. In mouse, we find no evidence of age-related progressive impairment of the oxidative phosphorylation system or increase of mitochondrial $H_2O_2$ production, whereas both effects are evident in *Drosophila*.

## Materials and methods

### Mouse breeding

All mice mutations in this study were on an inbred C57Bl/6N nuclear background. Mutator mice were generated as previously described (*Ross et al., 2013*). Briefly, $PolgA^{WT/mut}$ mice were generated by crossing a $PolgA^{WT/WT}$ female with a $PolgA^{WT/mut}$ male, and subsequently intercrossing the progeny to generate $PolgA^{mut/mut}$ (mutator mice). Mice were maintained on a standard mouse chow diet and sacrificed at different time points by cervical dislocation in strict accordance with the recommendations and guidelines of the Federation of the European Laboratory Animal Science Association (FELASA). Protocols were approved by the Landesamt für Natur, Umwelt und Verbraucherschutz, Nordrhein-Westfalen, Germany (Permit ref: 84–02.05.20.12.086).

### Fly breeding

$w^{Dah}$ wild-type flies were maintained at 25°C and fed a standard sugar/yeast/agar diet (SYA). Once mated, females, raised at controlled larval densities, were used. Adult flies were kept in SYA food vials containing 10 or 25 flies per vial for survival and climbing analysis, respectively. Climbing ability was assessed as previously described (*Greene et al., 2003*).

### Isolation of mouse mitochondria

Mice were sacrificed by cervical dislocation, and heart, liver and kidneys were quickly collected in ice-cold DPBS (Gibco), minced and homogenized with a few strokes of a Potter S homogenizer (Sartorius) in 5 ml (for heart and kidney) or 20 ml (for liver) of ice-cold mitochondria isolation buffer (MIB; 310 mM sucrose, 20 mM Tris-HCl, 1 mM EGTA, pH 7.2). Mitochondria were purified by differential centrifugation (1200 g for 10 min) and the supernatants were then centrifuged at 12,000 g for 10 min. The crude mitochondrial pellet was resuspended in an appropriate volume of MIB. Mitochondrial concentration was determined using the Protein DC Lawry based assay (Bio-Rad).

### Isolation of mitochondria from fruit flies

Fruit flies were homogenized with a few strokes of a loose Potter S homogenizer (Sartorius) in 5 ml of ice-cold mitochondria isolation buffer (MIB; 310 mM sucrose, 20 mM Tris-HCl, 1 mM EGTA, pH 7.2). After filtration through a 100 µm nylon mesh filter, mitochondria were further homogenized in a tight Potter S homogenizer (Sartorius) and purified by differential centrifugation (800 g for 10 min) and the supernatant was then centrifuged at 4500 g for 15 min. The crude mitochondrial pellet was resuspended in an appropriate volume of MIB. Mitochondrial concentration was determined using the Protein DC Lowry based assay (Bio-Rad).

### Mitochondrial respiratory assay

The rate of mitochondrial oxygen consumption was measured as previously described (*Freyer et al., 2012*) at 37°C using 65–125 µg of crude mitochondria in 2.1 ml of mitochondrial respiration buffer (120 mM sucrose, 50 mM KCl, 20 mM Tris-HCl, 4 mM $KH_2PO_4$, 2 mM $MgCl_2$, 1 mM EGTA, pH 7.2) in an Oxygraph-2k (Oroboros Instruments). Oxygen uptake was measured using either pyruvate/glutamate/malate (10 mM pyruvate, 5 mM glutamate and 5 mM malate) or 10 mM succinate and 10 nM

rotenone. Oxygen consumption was assessed under phosphorylating conditions with 1 mM ADP (state 3) or non-phosphorylating conditions by adding 2.5 µg/ml oligomycin (pseudo state 4). Respiratory control ratios (*Brand and Nicholls, 2011*) were above 10 with pyruvate/glutamate/malate and above 5 with succinate-rotenone. Respiration was uncoupled by successive addition of carbonyl cyanide m-chlorophenyl hydrazone (CCCP) up to 3 µM to reach maximal respiration. The same procedure was used for *D. melanogaster*, except that isolated mitochondria were first incubated with pyruvate/glutamate/malate/proline (10 mM pyruvate, 5 mM glutamate, 5 mM malate, 10 mM proline) for state 3, state 4 and the uncoupled state. Succinate (10 mM), glycerol-3-phosphate (10 mM) and finally rotenone (10 nM) were then added to determine the maximal respiration driven by succinate and glycerol-3-phospate versus complex I driven-respiration.

## Measurement of ATP synthesis flux (JATP)

Isolated mitochondria (65 µg/ml) were suspended in respiration buffer (see above). After addition of 2 mM succinate, 10 nM rotenone and 1 mM ADP, oxygen consumption and ATP synthesis rates were measured as previously described (*Mourier et al., 2010*). Aliquots were collected every 20 s and precipitated in 7% $HClO_4$/25 mM EDTA, centrifuged at 16,000 g for 10 min and then neutralized with 2 M KOH, 0.3 M MOPS. The ATP content in these samples was determined with the ATPlite 1step (PerkinElmer). In a parallel experiment, oligomycin (2.5 µg/ml protein) was added to the mitochondrial suspension to determine the rate of non-oxidative ATP synthesis.

## Measurement of reactive oxygen species

The rate of $H_2O_2$ production was determined by monitoring the fluorescence emission at 590 nm upon oxidation of the indicator dye Amplex Red (5 U/ml) in the presence of horseradish peroxidase (1 µM) with excitation at 560 nm. A standard curve was obtained by adding known amounts of $H_2O_2$ to the assay medium in the presence of the reactants. Mitochondria (65 µg protein $ml^{-1}$) were incubated in respiratory medium (see above) at 37°C. $H_2O_2$ production was initiated by substrate addition, and the rate was determined by monitoring the fluorescence change with time (*Votyakova and Reynolds, 2001*).

## Enzyme activities

Tissue proteins (15–50 µg) were diluted in phosphate buffer (50 mM $KH_2PO_4$, pH 7.4) followed by spectrophotometric analysis of isolated respiratory chain complex activities at 37°C using a Hitachi UV-3600 spectrophotometer. Citrate synthase activity was measured at 412 nm ($\varepsilon$ = 13,600 $M^{-1}cm^{-1}$) after addition of 0.1 mM acetyl-CoA, 0.5 mM oxaloacetate and 0.1 mM 5,5-dithiobis-2-nitrobenzoic acid (DTNB). NADH dehydrogenase activity was determined at 340 nm ($\varepsilon$ = 6220 $M^{-1}cm^{-1}$) after addition of 0.25 mM NADH, 0.25 mM decylubiquinone and 1 mM KCN and monitoring rotenone sensitivity. Succinate dehydrogenase (SDH) activity was measured at 600 nm ($\varepsilon$ = 21000 $M^{-1}cm^{-1}$) after addition of 40 mM succinate, 35 µM dichlorphenol indophenol (DCPIP) and 1 mM KCN. COX activity was assessed using a classical TMPD/ascorbate assay. Briefly, homogenized tissue (65 µg/ml) was suspended in mitochondrial respiration buffer (see above). Oxygen consumption was assessed in the presence of TMPD (0.2 mM), ascorbate (1 mM) and antimycin A (0.5 µM). After a few minutes of stationary respiration, KCN (2 mM) was injected into the chamber. COX activity corresponds to the KCN-sensitive respiration. Catalase activity was assessed using an Oroboros oxygraph. Catalase activity of homogenized tissues (65 µg/ml) was followed by recording the oxygen production in the presence of 0.01% $H_2O_2$. All chemicals were obtained from Sigma Aldrich.

## Western blot analysis

Proteins from tissue lysates were separated by SDS-PAGE and blotted onto PVDF membranes (GE Healthcare). The following primary antibodies were used: rabbit anti-superoxide dismutase 1 (1:1000, Abcam-ab16831) and rabbit monoclonal anti-superoxide dismutase 2 (1:1000, Millipore-06–984). The following HRP-conjugated secondary antibodies were used: donkey anti-rabbit IgG (Amersham, NA9340V) and sheep anti-mouse (Amersham, NXA931). For chemiluminescence detection, samples were incubated with ECL (GE Healthcare). Densitometry analyses were performed using the FIJI software.

## Electron cryo-tomography

Mitochondria were washed twice with 320 mM trehalose, 20 mM Tris pH 7.3, and 1 mM EGTA. Samples were mixed 1:1 with fiducial gold markers (10 nm gold particles conjugated to protein A, Aurion, The Netherlands) and immediately plunge-frozen in liquid ethane on Quantifoil holey carbon grids (Quantifoil Micro Tools, Germany). Single tilt series (typically ±60°, step size 1–1.5°) were collected at 300 kV with an FEI Polara electron microscope equipped with a post-column Quantum energy filter and an Ultrascan 4 × 4 k CCD camera (Gatan, USA), or with a post-column Tridiem energy filter and a 2 × 2 k CCD camera (Gatan). Alternatively, tilt series were recorded with an FEI Titan Krios electron microscope equipped with a Quantum energy filter and a K2 summit direct electron detector (Gatan). Underfocus was 8–9 μm and the magnification was chosen to give an object pixel size between 4.3 Å and 7.3 Å. The total electron dose per tilt series was 120–150 e⁻/Å². Tilt series were aligned to gold fiducial markers and tomograms were reconstructed by back-projection with the IMOD software package (*Kremer et al., 1996*). A final filtering step applying non-linear anisotropic diffusion (*Frangakis and Hegerl, 2001*) was performed to increase contrast. Tomograms were manually segmented with the program AMIRA (FEI).

## Acknowledgements

We acknowledge funding from the Max Planck Society and the Bundesministerium für Bildung und Forschung (SyBACol 0315893A-B) (to LST). NGL received additional support from the Swedish Research Council (2015–00418) and the Knut and Alice Wallenberg Foundation.

## Additional information

### Competing interests
WK: Reviewing editor, *eLife*. The other authors declare that no competing interests exist.

### Funding

| Funder | Author |
| --- | --- |
| Max-Planck-Gesellschaft | Werner Kühlbrandt |
| Deutsche Forschungsgemeinschaft | Werner Kühlbrandt |

The funders had no role in study design, data collection and interpretation, or the decision to submit the work for publication.

### Author contributions
TB, Performed and analyzed experiments, interpreted the data, and wrote the paper; AM, Performed and analyzed experiments; LST, Performed and analyzed experiments and provided materials; LP, Provided materials; N-GL, Initiated and supervised the project and provided materials; WK, Conceptualization, Resources, Data curation, Supervision, Funding acquisition, Investigation, Methodology, Writing—original draft, Project administration, Writing—review and editing, Initiated and supervised the project, interpreted the data and wrote the paper

### Author ORCIDs
Werner Kühlbrandt, http://orcid.org/0000-0002-2013-4810

### Ethics
Animal experimentation: This study was performed in strict accordance with the recommendations and guidelines of the Federation of European Laboratory Animal Science Associations (FELASA). The protocol was approved by the Landesamt für Natur, Umwelt und Verbraucherschutz in Nordrhein-Westfalen, in Germany (Permit ref: 84-02.05.20.12.086).

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
