## [Decision Letter]

Thank you for submitting your article "Changes of mitochondrial ultrastructure and function during ageing in mice and *Drosophila*" for consideration by *eLife*. Your article has been reviewed by three peer reviewers, and the evaluation has been overseen by Sriram Subramaniam as Reviewing Editor and Richard Aldrich as the Senior Editor. The following individuals involved in review of your submission have agreed to reveal their identity: Luigi Ferrucci and Terrence Frey.

The reviewers have discussed the reviews with one another and the Reviewing Editor has drafted this decision to help you prepare a revised submission.

Summary:

Brandt et al., present a thorough study of the effects of aging in two model systems, mice (including mtDNA mutator mice) and *Drosophila melanogaster* that combines careful measurements of critical mitochondrial function, enzyme content, and the production of damaging reactive oxygen species (in the form of hydrogen peroxide) on purified mitochondria with state of the art cryo-electron tomorgraphy to elucidate changes in inner membrane topography. The goal is to investigate the hypothesis that aging is manifest in cellular mitochondria and mediated by increased production of reactive oxygen species. The study includes comparison of aging on the structures of mitochondria in three critical mouse tissues; heart, liver and kidney. *Drosophila* mitochondria were isolated from whole organisms. Except for the aged *Drosophila* mitochondria, the physiological measurements and enzyme contents do not change dramatically (or in many cases even significantly) with aging, but there are dramatic ultrastructural changes. Overall, the study combines careful measurements of critical mitochondrial function, enzyme content, and the production of damaging reactive oxygen species (in the form of hydrogen peroxide) on purified mitochondria with state of the art cryo-electron tomorgraphy to elucidate changes in inner membrane topography.

Essential revisions:

1) Some additional references to include: Sun et al., 2007 who studied inner membrane remodeling extensively in a HeLa cell culture model of apoptosis that is relevant to the changes observed in this paper. These changes were also observed by Merkwirth et al., 2008 who manipulated the protein OPA1 that regulates crista junction structure.

2) One of the reviewers appeared surprised that the authors did not detect a reticular structure of mitochondria network in the heart data, since this type of structure has been identified in other studies, or substantial differences between the sub-sarcolemmal and inter-myofibrillar mitochondria, since clear morphological differences have been identified with much less sophisticated imaging. The authors should comment on whether such structures may have been present but missed, given the isolation techniques used.

3) The analysis of the tomographic data is largely descriptive, with percentages given for the number of mitochondria falling into each category. In the majority of places there is no indication of the number of mitochondria considered, nor the number of animals from which they came. This makes the percentages hard to assess. The authors should provide a table detailing the number of animals and the number of mitochondria analyzed for each condition. Extracting as many quantitative parameters as possible to describe the morphological phenotypes would also be a good idea. Adjectives such as "several", "some", etc. should be replaced by numbers wherever possible. Additionally, the authors should clarify what fraction of inner membrane is lost to the "voids". Circular or slightly elongated crista junctions are mentioned but without sizes specified. Please provide measurements of crista junction sizes of a representative sample.

4) Since purified mitochondria are being analyzed, discussion of possible changes in structure during purification should be included. Also, in most of the tomograms, the cristae look fine, but in the young mouse liver, the cristae seem enlarged (bloated) and the matrix dense as if the mitochondrion is in a condensed state.

---

## [Author Response]

*Essential revisions:*

*1) Some additional references to include: Sun et al., 2007 who studied inner membrane remodeling extensively in a HeLa cell culture model of apoptosis that is relevant to the changes observed in this paper. These changes were also observed by Merkwirth et al., 2008 who manipulated the protein OPA1 that regulates crista junction structure.*

Thank you for bringing this point to our attention. We have included these citations in subsection “Tomography of mouse kidney mitochondria” and the Discussion section of the revised manuscript, and have added a sentence referring to the OPA1 mutants in subsection “Tomography of mouse kidney mitochondria”.

*2) One of the reviewers appeared surprised that the authors did not detect a reticular structure of mitochondria network in the heart data, since this type of structure has been identified in other studies, or substantial differences between the sub-sarcolemmal and inter-myofibrillar mitochondria, since clear morphological differences have been identified with much less sophisticated imaging. The authors should comment on whether such structures may have been present but missed, given the isolation techniques used.*

The reviewer may be thinking of plastic sections of cardiac tissue, where mitochondrial networks in the sarcolemma or in myofibrils may indeed look substantially different. We would like to point out that cryo-tomography of isolated mitochondria cannot visualize intact mitochondrial networks, because they are necessarily disrupted during isolation. The isolated mitochondria from both cell types would look very similar, even if the networks may look different in plastic sections.

*3) The analysis of the tomographic data is largely descriptive, with percentages given for the number of mitochondria falling into each category. In the majority of places there is no indication of the number of mitochondria considered, nor the number of animals from which they came. This makes the percentages hard to assess. The authors should provide a table detailing the number of animals and the number of mitochondria analyzed for each condition.*

Thank you for this suggestion. We have now added such a table (Table 1).

*Extracting as many quantitative parameters as possible to describe the morphological phenotypes would also be a good idea.*

Done, see Table 1.

*Adjectives such as "several", "some", etc. should be replaced by numbers wherever possible.*

The adjectives have been replaced by exact numbers and standard deviations wherever possible.

*Additionally, the authors should clarify what fraction of inner membrane is lost to the "voids".*

This information is now included in a paragraph in subsection “Tomography of mouse liver mitochondria” of the revised manuscript.

*Circular or slightly elongated crista junctions are mentioned but without sizes specified. Please provide measurements of crista junction sizes of a representative sample.*

These measurements are now included in the first paragraph in subsection “Tomography of mouse heart mitochondria” of the revised manuscript.

*4) Since purified mitochondria are being analyzed, discussion of possible changes in structure during purification should be included. Also, in most of the tomograms, the cristae look fine, but in the young mouse liver, the cristae seem enlarged (bloated) and the matrix dense as if the mitochondrion is in a condensed state.*

As far as possible, mitochondria from different organisms and tissues were treated in the same way. The fact that there are clear differences e.g. between young and old mouse liver, and that these look different from heart and kidney mitochondria allows us to conclude that the isolation process itself does not affect mitochondrial membrane structure significantly. We can further conclude that the different appearance of the young mouse liver mitochondria is not an artefact of isolation, but a genuine feature. These points are now discussed in the first paragraph in the Discussion of the revised manuscript.